# Efficient Human Violence Recognition for Surveillance in Real Time

**DOI:** 10.3390/s24020668

**Published:** 2024-01-20

**Authors:** Herwin Alayn Huillcen Baca, Flor de Luz Palomino Valdivia, Juan Carlos Gutierrez Caceres

**Affiliations:** 1Academic Department of Engineering and Information Technology, Professional School of Systems Engineering, Faculty of Engineering, Jose Maria Arguedas National University, Andahuaylas 03701, Peru; fpalomino@unajma.edu.pe; 2Academic Department of Systems and Informatics Engineering, Professional School of Computer Science, Faculty of Production and Services Engineering, San Agustin of Arequipa National University, Arequipa 04001, Peru; jgutierrezca@unsa.edu.pe

**Keywords:** human violence recognition, video surveillance, real time, spatial attention, spatial motion extractor, short temporal extractor, global temporal extractor, VioPeru

## Abstract

Human violence recognition is an area of great interest in the scientific community due to its broad spectrum of applications, especially in video surveillance systems, because detecting violence in real time can prevent criminal acts and save lives. The majority of existing proposals and studies focus on result precision, neglecting efficiency and practical implementations. Thus, in this work, we propose a model that is effective and efficient in recognizing human violence in real time. The proposed model consists of three modules: the Spatial Motion Extractor (SME) module, which extracts regions of interest from a frame; the Short Temporal Extractor (STE) module, which extracts temporal characteristics of rapid movements; and the Global Temporal Extractor (GTE) module, which is responsible for identifying long-lasting temporal features and fine-tuning the model. The proposal was evaluated for its efficiency, effectiveness, and ability to operate in real time. The results obtained on the Hockey, Movies, and RWF-2000 datasets demonstrated that this approach is highly efficient compared to various alternatives. In addition, the VioPeru dataset was created, which contains violent and non-violent videos captured by real video surveillance cameras in Peru, to validate the real-time applicability of the model. When tested on this dataset, the effectiveness of our model was superior to the best existing models.

## 1. Introduction

In recent years, with the development of real-time video platforms and video cameras, the availability of visual data has increased rapidly. Due to this constant growth, computing has needed to keep pace in terms of providing users with well-organized information and, above all, valuable services. To achieve this, video processing and analysis focuses on identifying patterns in the data to enhance the aspects that are widely used in modern society [1].

Video processing and analysis has been a topic of interest in machine learning and pattern recognition for years. It focuses on many different problems and tasks, such as action recognition [2], action localization [3], anomaly detection [4], and scene recognition [5], among others.

One of the main difficulties when processing videos is their high spatiotemporal nature. Each frame, in principle, can be viewed as a static image containing visual (spatial) information. This simple fact makes the video processing task computationally expensive, even when processing a short video clip, since it can include many images; furthermore, because there is a dynamic component between the spatial content of consecutive frames, a temporal dimension is created.

The question of how to describe spatial and temporal information in order to understand the content of a video continues to be a challenge, especially when trying to provide models that can be applied to real problems.

Efficient human violence recognition is an area of great interest due to its wide range of applications, for example, in robotics, medicine, psychology, human–computer interaction, and video surveillance. An automatic system for detecting human actions and violence would be able to send an alert concerning a certain incident or crime and allow measures to be taken to mitigate it. To achieve this, it is essential to detect violent activity in real time. Although violence recognition in video surveillance has significantly improved, most studies aim to improve accuracy on known datasets rather than exploring real-world scenarios.

There are many techniques for detecting violence in videos. Typical methods utilize optical flow as the input [6,7,8,9,10]. In addition, two-stream, CNN variants, and 3D CNN variants have achieved good results by combining optical flow with other inputs such as RGB frames [11,12,13,14]. Optical flow is a motion representation for video action recognition tasks. However, extracting optical flow incurs a significant computational cost and is inefficient for real-time violent human action recognition tasks.

The most promising techniques are based on deep learning [13,15,16,17,18,19], which, rather than optical flow, use neural networks for feature extraction, encoding, and classification. These techniques achieve a better performance, reducing the computational cost as compared to those that use optical flow. However, they are still costly in terms of parameters and FLOPS, and so applying them in a real-world scenarios remains a challenge.

Herein, we focus on recognizing violent human actions in video surveillance that can be applied in real-world scenarios. In this context, classification models must identify human violence at the precise moment of its occurrence, that is, in real time.

To the best of our knowledge, there are no datasets specifically aimed at the video surveillance domain; the current reference datasets contain a mixture of videos taken from mobile devices, movies, and hockey, where the camera adopts characteristics and positions oriented toward the best shot; however, in a real-world scenario, this does not happen. The violent human actions captured by video surveillance cameras are more complex. Various factors often act to degrade the scene, such as occlusion between the individuals, the time of day, excessive artificial light from poles and/or vehicles, the type of camera, the camera resolution, and the size of the violent scene as compared to the size of the frame. In fact, in a recent review, these aspects were highlighted as challenges that need to be overcome by the currently proposed models [20], which are mainly oriented towards effectiveness when using known datasets.

Thus, herein, an efficient and effective model based on deep learning is proposed for the recognition of violent human actions in real-time video surveillance, which can be used in real-world scenarios.

### 1.1. Problem

Current proposals for the recognition of violent human actions solely focus on improving effectiveness using general datasets, which are not oriented towards a specific domain, and neglect applicability in real-world scenarios. Thus, models that are efficient enough to be deployed in devices used in real-time video surveillance, with cutting-edge effectiveness, whose results have been tested on datasets that consider the real challenges of video surveillance are required.

In a recent study by Ullah et al. [20], it was observed that, in the field of human action recognition, there are still challenges that are not addressed by current proposals. These challenges include occlusion, differences in indoor and outdoor cameras, lighting variation in different scenarios, scenes involving crowds, real-time processing, and the complexity and efficiency of existing approaches.

In the specific context of violence detection in video surveillance, additional problems can be identified, such as the proportion of the violent action in relation to the size of the video frame, the direction of focus of the violent action, and the differences between day and night. To our knowledge, no proposal has comprehensively addressed these issues; instead, the primary focus has been on improving the effectiveness of the models on specific datasets. It is important to note that current datasets are generic and heterogeneous, which underlines the need to consider the above-mentioned issues and develop datasets that are more oriented towards specific domains.

### 1.2. Motivation

This study was motivated by the fact that although there are different proposals for the recognition of violent human actions in videos, there is no definite proposal aimed at solving the identified problems, i.e., a model that can be used in a real scenario and is both efficient and effective.

The majority of proposals focus on effectiveness and few focus on efficiency. Thus, there exist very accurate models, but they are complex with high computational costs and cannot be used in real time.

In summary, the inefficiency of the proposals in real applications, their limited applicability to a specific domain, and the aforementioned underlying challenges constitute the motivation of this work.

### 1.3. Objectives

The general objective of this work was to propose a model based on deep learning for recognizing violent human actions in real-time video surveillance.

Our specific objectives were as follows:Collect videos from real surveillance cameras that integrate characteristics of human violence and later publish them in a dataset;Develop an accurate model based on attention and temporal fusion mechanisms;Develop a model based on temporal changes and 2D CNN that is efficient in terms of the number of parameters and FLOPS;Develop a compact model for recognizing violent human actions in video surveillance with minimal latency, i.e., close to real time.

### 1.4. Contributions

The contributions of this study are as follows:A model for recognizing violent human actions that can be used in real-world scenarios in real time;An accurate and effective model for recognizing violent human actions that is efficient in terms of the number of parameters and FLOPS, the results of which contribute to the state of the art;A published dataset oriented toward the domain of video surveillance.

### 1.5. Work Organization

Section 2 describes the techniques and results of the best proposals related to the objectives of this study. Section 3 first describes the work related to our proposal and subsequently details and explains the operation of the proposed architecture and its respective modules. Section 4 presents the results according to our objectives. Finally, Section 6 presents the conclusions of this work.

## 2. Related Work

Human action recognition is an area of active research, and therein, there exist many approaches; in this section, we present some of them, starting with more straightforward approaches and ending with the most novel contributions. First, the work related to the proposed modules is explored, and then, 2D CNN-based models related to the backbone of our proposal are addressed. Finally, a review of the cutting-edge techniques for recognizing violent human actions in video surveillance, considering the Movies [21], Hockey [21], and RWF-2000 dataset [22], is presented.

### 2.1. Work Related to the Proposal

#### 2.1.1. Extraction of Regions of Interest

Video surveillance cameras are mostly fixed-position cameras, and violent scenes do not generally occupy the entire area (HxW) of the video. On the contrary, the scenes usually occupy a small portion of the video; in this way, the remaining area becomes the background of the violent scene, whose spatiotemporal characteristics do not contribute to recognition efficiency, but rather act to reduce it. On the other hand, extracting these redundant features makes the models less efficient. Therefore, extracting the regions of interest from each frame is essential for effectiveness and efficiency.

Extracting regions of interest has been addressed using attention mechanisms. Ulutan et al. [23] proposed to extract regions of moving actors; for this, they used object detectors with an I3D architecture as a backbone. They also used the amplification and attenuation of the actors. Amplification and attenuation are essential in the extraction of regions of interest. Our proposal carries out the same process without detectors but with morphological deformation processes in each frame. This improves the model’s efficiency, as it no longer uses detectors based on pretrained backbones.

When addressing the inefficiency of optical flow in identifying movement limits, Zhang et al. [24] presented a proposal based on the Euclidean distance of two consecutive frames before using a convolution backbone. Their proposal is efficient and significantly reduces the number of FLOPs in the process; however, the use of the backbone is still heavy in terms of efficiency. We took this proposal as a reference, using the Euclidean distance of two consecutive frames but without the convolution backbone. Instead, we use morphological deformations to represent the regions of interest.

#### 2.1.2. Short-Duration Spatiotemporal Feature Extraction

Proposals based on 3D CNN networks, such as those proposed by Tran et al. [13] and Carreira et al. [25], can simultaneously extract spatiotemporal features. However, they incur a high computational cost. To address this problem, several proposals replace 3D CNN networks with 2D CNN networks without compromising effectiveness and improving efficiency.

Lee et al. [26] proposed to extract spatiotemporal features from motion filters in a 2D CNN network, and Xie et al. [27] proposed to mix modules based on 3D CNN and 2D CNN networks. Both studies achieved adequate effectiveness results using human action recognition datasets, but without using violence-oriented datasets. These proposals are considered in our work, as using a 2D CNN network generates better efficiency conditions, especially if the objective is detection in real time.

In this way, Lin et al. [28] proposed to use 2D CNN networks but with a substantial improvement, i.e., a temporal change module, in which consecutive frames replace the dimension of the channels of the frames and carry out the extraction of spatiotemporal features with several convolutions. Our proposal considers replacing channel information with temporal information from consecutive frames, but we only use a single 2D CNN network (backbone). Finally, in this case, it is possible to extract short-duration spatiotemporal characteristics from three consecutive frames, significantly reducing the number of FLOPs as compared to Lin et al.’s proposal [28].

#### 2.1.3. Global Spatiotemporal Feature Extraction

Extracting spatiotemporal features from three consecutive frames can be applied to recognize human actions, as was proposed by Huillcen et al. [29]. This method produces better efficiency results but still fails to surpass the state-of-the-art proposals in terms of effectiveness. Our proposal adds a module to extract temporal characteristics from a larger set of frames, for example, 30. In this way, our model searches for movement characteristics over more frames (not just three) without compromising efficiency.

The proposal of Zhang et al. [24] has a feature reduction module in the time dimension using max-pooling layers. This is merged with its features and can enable the recognition of human actions. Our proposal takes this module, but it only uses two average pooling layers, which act to ensure the compactness of the model, merge it with the input, and recognize violent actions through fully connected layers.

### 2.2. Reference Backbones

As described in the previous section, we considered using a pretrained 2D CNN backbone. The different alternatives were compared in order to choose the backbone used in our proposal.

In the literature review, different proposals and techniques stand out. We considered efficient and effective models. These included ResNet50 [30], InceptionV3 [31], DenseNet121 [32], SqueezeNet [33], MobileNet V2 [34], MobileNet V3 [35], EfficientNet B0 [36], MnasNet [37], GhostNet V2 [38], and vision transformers [39].

The comparative analysis was performed based on the effectiveness results using the ImageNet dataset [40]. This original dataset has 1,280,000 training images and 50,000 validation images with 1000 classes.

Table 1 shows a comparative analysis of these models. The possible candidates are marked in bold.

Vision transformers exhibit good performance in sequence-based problems [41], especially in natural language processing tasks, and image detection and recognition tasks [39]. Similarly, they have been used in applications for recognizing violent actions [42]. However, to our knowledge, they have yet to be evaluated on the RWF-2000 reference dataset [22], nor are there any results on their efficiency. We rejected this model for this reason and because, in general, proposals based on transformers aim to improve the effectiveness of their results but at a higher computational cost than proposals based on 2D CNNs.

### 2.3. Benchmark

#### 2.3.1. Benchmark on Classic Datasets

The methods for recognizing violent human actions are divided into two groups: handcraft methods and deep learning methods. Handcraft methods do not achieve good results, especially in terms of efficiency. The most representative works in this group include that of Gao et al. [6], with Oriented Violence Flows (OViF); Deniz et al. [7], with Radon transform; Bilinski et al. [8], with Fisher vectors; Zhang et al. [9], with the Weber Local Descriptor (MoI-WLD); and Deb et al. [10], with Outlier-Resistant VLAD (OR-VLAD).

Deep learning methods use deep neural networks as feature extractors. The most important include Dong et al. [15], with multiple streams based on the stream model [12]; Zhou et al. [16], with Time Slice Networks (TSN) and FightNet; and Serrano et al. [17], with Hough forests. These proposals still use optical flow combined with deep learning; therefore, they still have efficiency problems and dependence on handcraft methods.

These problems are addressed by proposals such as that of Sudhakaran et al. [18], with 2D ConvNets and ConvLSTM, and Hanson et al. [19], with the ConvLSTM (Bi-ConvLSTM) architecture.

Recently, 3D CNN-based approaches have improved the effectiveness of previous proposals but at a high computational cost, which is typical of 3D CNN models, for example, the proposals of Tran et al. [13] and Li et al. [43].

An improvement to the previous approach in terms of efficiency was presented by Huillcen et al. [44]. It uses a DenseNet architecture but with different configurations of dense layers and dense blocks to ensure the compactness of the model. Later, Huillcen et al. [29] presented a new proposal based on extracting spatiotemporal features using a 2D CNN and extracting regions of interest to ensure model compactness.

Table 2 summarizes the results of all these proposals in classic datasets: the Hockey Fights dataset [21], the Movies dataset [21], and the Violent Flows dataset [45].

#### 2.3.2. Benchmark on RWF-2000 Dataset

The methods described in the previous subsection used analyses on datasets that are not specific to video surveillance. In addition, no analysis of efficiency was performed. Thus, these approaches are oriented towards effectiveness but not efficiency.

An RWF-2000 dataset was recently proposed by M. Cheng et al. [22], which consists of 2000 videos extracted from YouTube. These videos are of different resolutions, sources, and camera positions, which makes it a reference dataset and a challenge for methods tested using classic datasets. Below, there is an analysis of the recent proposals tested on this dataset, with effectiveness and efficiency results.

According to a recent study by Mumtaz et al. [46], the main studies for recognizing violent human actions contain little information about efficiency results, i.e., information about the complexity and number of parameters, confirming that their objective was to find good results in terms of accuracy while increasing the complexity of the model. It was shown that proposals based on optical flow, 3D CNN, LSTM, two-stream networks, and 3D skeletons have high computational costs and cannot be used in real-time scenarios.

However, there are studies whose objectives were to find models with good effectiveness and efficiency results (see Table 3 and Figure 1). These include Carreira et al. [25], with the I3D’s feature-based mechanism, and Cheng et al. [22], where the authors achieved an accuracy of 87.25% with only 0.27 million parameters; however, in the complexity calculation, they did not consider preprocessing with optical flow.

Sudhakaran et al. [18] applied 2D ConvNets to extract spatial feature maps, followed by ConvLSTM to encode the spatiotemporal information, producing an efficiency of 77%, but at a high cost (94.8 million parameters). Something similar happened in Tran et al. [13], with their proposal based on 3D CNN.

There are techniques based on 3D skeletons, the most representative of which was presented by Su et al. [48], who achieved an efficiency of 89.3%; however, using the extraction of key points in the recognition of skeletons has several associated problems. These include a high computational cost and an unsuitability for the domain in video surveillance, because in a real-world scenario, there is no camera focus towards the violent scene.

Outstanding proposals also emerged based on a spatial and temporal feature extraction sequence. This reduced the complexity of 3D CNN networks by taking advantage of the efficiencies of 2D CNNs. Huillcen et al. [29] improved their proposal in terms of efficiency but without surpassing the effectiveness of other proposals. For this, their model was based on replacing the 3D CNN approach with a 2D CNN and using preprocessing to identify regions of interest.

To the best of our knowledge, the best proposal in terms of efficiency and effectiveness on the RWF-2000 dataset [22] was presented by Islam et al. [47]. It is based on a separable convolutional network (SepConvLSTM) and MobileNet and reached an efficiency of 89.75% with 0.333 million parameters and 1.93 GFLOPs.

However, the efficiency results are questionable since MobileNet V2 [34] alone has 3.4 million parameters.

Analyzing the architecture presented by Islam et al. [47] (See Figure 2), we found that it is a two-stream CNN-LSTM model. As can be seen, each flow has a MobileNet V2 backbone [34], and according to Table 1, each MobileNet V2 backbone [34] has 3.4 million parameters. Islam et al. [47] used two MobileNet v2 backbones [34] and then fused them with a SepConvLSTM layer. Thus, it is a questionable claim that their proposal only has 0.333 million parameters. In reality, it should have more than 3.4 million parameters as a result of using two MobileNet V2 backbones [34] and adding the parameters of the SepConvLSTM layers. The same applies for the FLOPs analysis.

## 3. Proposal

According to the review of the state of the art and in response to the challenges of proposing an efficient, effective model that can be used in real time, we propose an architecture composed of three modules: a Spatial Motion Extractor (SME) module, which functions to extract regions of interest from a frame; a Short Temporal Extractor (STE) module, whose function is to extract temporal characteristics of short-duration fast movements; and a Global Temporal Extractor (GTE) module, which identifies long-term temporal features and fine-tunes the model for better accuracy.

### 3.1. Proposal Architecture

The general objective of our proposal has an inverse nature: to be efficient and, at the same time, effective at recognizing human actions in real time. When reviewing the state of the art, it can be seen that high-efficiency proposals often have low efficiency, and vice versa. In this way, the proposal contains modules that improve effectiveness using efficient methods. Thus, a hybrid architecture is proposed, which is composed of three techniques: spatial motion extraction, a 2D CNN with frame averaging, and temporal feature extractors. Figure 3 shows the proposed architecture.

The Spatial Motion Extractor (EME) module takes the 30 resized video frames as input and extracts a region of interest from each frame. This region corresponds to the violent scene. The extraction process first uses the Euclidean distance of two consecutive frames to extract the movement limits. They are dilated with morphological deformations, and finally, the dot product is produced between the result and the second frame, obtaining the frame with the violent scene highlighted on a black background. Unlike other proposals that use neural networks, transfer learning, or a sequence of convolutions, ours behaves as an attention mechanism that only uses image processing operations. This module is crucial to the model’s efficiency since the region of interest is extracted at a cost of 0 parameters.

The Short Temporal Extractor (STE) module takes the frame with the violent scene highlighted and extracts fast-moving spatiotemporal features, specifically from three consecutive frames. This process uses the MobileNet V2 backbone as a spatial feature extractor. To extract temporal features, a three-channel frame is assembled, where each channel is the average of the RGB channels of the three consecutive frames. It is possible to extract spatiotemporal characteristics with a 2D CNN backbone. This process is vital to the model’s efficiency since it reduces the input frames from 30 to 10, significantly reducing the FLOPs when deployed in a real-world scenario. In addition, it takes advantage of the efficiency of the 2D CNN backbone, which is superior to classic alternatives based on 3D CNN, LSTM, two stream, and others that increase the complexity of their proposals.

The Global Temporal Extractor (GTE) module takes the 10 spatiotemporal features from the previous module and fine-tunes the model. The process uses AVG pooling and fully connected layers to extract motion features from the 10 frames, i.e., from long-duration movements. In this manner, it is possible to increase the model’s effectiveness without compromising too much on efficiency. This module is crucial and improves the effectiveness of the model without compromising too much on efficiency.

The input is the sequence of video frames $F_{t},F_{t+1},F_{t+2},\ldots ,F_{t+T}$, for 1≤t≤T, and T=30, which are resized to a resolution of 224 × 224 pixels. The details of each module are detailed in the subsequent subsections.

### 3.2. Spatial Motion Extractor (SME) Module

This module is based on the natural process of a human being when observing a scene. When the scene is static, sensory attention covers the entire scene; however, when some movement occurs, sensory attention is oriented toward the specific movement area, making visual perception and the possible identification of the movement more successful. We took this natural process into account when designing the frame-by-frame preprocessing of the videos in order to extract one or several regions of interest. These are based around movement since all violent human action contains this feature, especially rapid movement.

Extracting a region of the frame where movement occurs increases recognition efficiency since it forces the model to only extract characteristics from the movement, leaving aside areas that do not contribute to or reduce the effectiveness. This module is essential to our proposal. The details are shown in Figure 4.

The Spatial Motion Extractor (SME) module takes two consecutive RGB frames $F_{t},F_{t+1}\in R^{3\times W\times H}$ for 1≤t≤T and calculates the Euclidean distance *D* for each pixel and each channel according to:(1)$$d_{t}=\sqrt{\sum_{i=1}^{3}{\left(F_{t+1}^{i}-F_{t}^{i}\right)}^{2}}$$
where *T* represents the number of frames to be processed, *i* represents the RGB channels, *F* is a specific frame, and $d_{t}\in R^{3\times W\times H}$. When a pixel remains at the same value and position, the difference is zero. Therefore, the pixels without movement are black, causing the resulting frame to extract the background of the initial frame, but if the pixel in the same position changes value, there is some movement, and the difference will be one grayscale pixel. Figure 5 shows an example of the administrative distance between two consecutive frames:

It is observed that $d_{t}$ represents the grayscale motion limits of the frames $F_{t},F_{t+1}$ with the respective removal of the background. However, $d_{t}$ does not yet represent a region of interest. Morphological deformations are applied so that the limits of movement act as perimeters of the region of interest, in such a way as to convert limits into regions. For this, dilation is used with a 3 × 3 kernel and 12 iterations. Finally, Figure 6 shows the result $b_{t}$ of the previous example.

With $b_{t}$, the region is identified, but the actual pixels of the movement are not; thus, with a dot product procedure between the frame $F_{t+1}$ and $b_{t}$, we obtain $M_{t}\in R^{3\times W\times H}$ (see Figure 7), which represents the region of interest where movement occurs, with the elimination of the background.

### 3.3. Short Temporal Extractor (STE) Module

This module has the function of improving effectiveness and efficiency in recognizing violent human actions. To do this, we consider that violent actions, such as punching, kicking, throwing, and others, are rapid movements, which can be reflected in the variation of pixels in consecutive frames. Thus, it is proposed to extract short-duration spatiotemporal characteristics, specifically from three consecutive frames.

Extracting spatiotemporal features is the fundamental task of recognizing human actions in general. Many techniques have been proposed for this task in recent years. As discussed in Section 2, 3D CNN-based architectures are the most appropriate to extract these spatiotemporal features. However, given their high computational complexity as a result of the number of parameters and FLOPS, they are unfeasible for real-time video surveillance. Although there are many other techniques to reduce the associated computational cost, to the best of our knowledge, there is currently no model for recognizing human actions in a real-world scenario.

According to the above, the Short Temporal Extractor (STE) module takes three consecutive RGB frames from the Spatial Motion Extractor (SME) module $M_{t}$, $M_{t+1}$, and $M_{t+2}$, transforms them into a single frame $P_{t,t+1,t+2}$, and finally extracts spatiotemporal features through a 2D CNN network. This constitutes the backbone of this module. We selected the MobileNet V2 network [34] for this. Figure 8 shows the details of this module.

In the above figure, $p_{t}$ represents the average of the three channels *c* of $M_{t}$, according to:(2)$$p_{t}=\sum_{c=1}^{3}\left(M_{t}^{c}\right)$$

In this way, when $p_{t+1}$ and $p_{t+2}$ are obtained, they are considered as channels and are assembled into a simple frame $P_{t,t+1,t+2}$, as if it were an image. Moreover, the color information of the boxes $M_{t}$, $M_{t+1}$, and $M_{t+2}$ is lost, which does not contribute to the recognition of violent human actions, but the extraction of short-term temporal characteristics is possible.

On the other hand, the Short Temporal Extractor (STE) module improves the efficiency of the model since, as can be seen, the number of processed frames *T* is reduced to a third: $\frac{T}{3}$, substantially reducing the number of FLOPs during processing in the 2D CNN network.

The frame $P_{t,t+1,t+2}$ containing temporal information enters a 2D CNN network, as if it were an image, to extract spatiotemporal features. The 2D CNN network chosen was MobileNet V2 [34] because it is the most efficient and produces the best results. The output is a feature map $B\in R^{\frac{T}{3}\times C\times W\times H}$, where *C* is the number of channels and H,W is the size of a feature matrix.

### 3.4. Global Temporal Extractor (GTE)

The Short Temporal Extractor (STE) module can capture spatiotemporal features *B* and, without any additional module, recognize violent human actions, as proposed by Huillcen et al. [29]. However, it does not take into account temporal characteristics between all frames $\frac{T}{3}$. In this way, it is possible to improve efficiency with some modifications that do not compromise effectiveness too much.

The feature map *B* contains information from the frames $\frac{T}{3}$ and each channel frame *C*. Considering this, by processing the relationship between the channels of each frame, global spatiotemporal characteristics are obtained that improve the effectiveness of the model, as in the proposal of Zhang et al. [24]. Here, a reduction process is performed, where we reduce the dimensions of *B* spatially and temporally, and then merge the features with a final fully connected layer to obtain the outputs “violence” and “no violence”. Figure 9 shows the Global Temporal Extractor (ETG) module in a general way.

Taking the feature map *B* as input, we perform a spatial compression or reduction process to obtain *S* in the form:(3)$$S^{c}=\frac{1}{H\times W}\sum_{i=1}^{H}\sum_{j=1}^{W}B^{c}\left(i,j\right)$$
where H×W is the size of a channel, and $S^{c}$ represents the average of the elements of the channel *c*, corresponding to applying a Global Average Pooling layer. The result *S* is a set of vectors of the form $S^{c}=\left[b_{1}^{c},b_{2}^{c},\ldots ,b_{\frac{T}{3}}^{c}\right]$.

Once again, a temporal compression process is carried out on the set of vectors *S* for each channel to obtain *Q* in the form:(4)$$q=\frac{1}{C}\sum_{c=1}^{C}S^{c}$$
where *q* is a vector $q=[q_{1},q_{2},\ldots ,q_{\frac{T}{3}}]$, which connects two fully connected layers followed by a sigmoid function and obtains the temporal characteristics *E*, which, in turn, is a vector $\left[e_{1},e_{2},\ldots ,e_{\frac{T}{3}}\right]$.

The next step is to recalibrate the weights (excitation) and obtain the characteristics resulting from the channel relationships. To do this, a point-to-point multiplication is performed between *E* and *S*. The result is added over time, and a new vector is obtained, representing the final temporal characteristics. Therefore, it is connected to the fully connected layer. The final output is the recognition of “violence” and “non-violence”.

## 4. Results

We show the results of the proposal according to its efficiency, effectiveness, and ability in real time. But first, the dataset and the model configuration are described.

### 4.1. Datasets

According to the Section 2, there are several freely distributed datasets; however, only some were taken as a reference to evaluate the performance of the different proposals. For our tests, we used the classic datasets Hockey Fights [21], Movies [21], and the reference dataset RWF-2000 [22].

As a contribution, we present the VioPeru dataset, created with real sources from Peruvian video surveillance cameras.

#### 4.1.1. Hockey Fights Dataset

The Hockey Fights [21] dataset contains 1000 clips extracted from hockey games. Figure 10 shows an example of a clip classified as Fight (Fi).

This dataset presents sequences of hockey fights. These are usually two-person fights, and the individuals are almost always wearing the same sports clothing. The violent action occupies almost the entire frame, and the background is very similar in all videos. These characteristics differ from a violent scene recorded by a video surveillance camera. Therefore, there are better candidates than this dataset to train a human action recognition model in video surveillance. However, we used it as a reference to compare the effectiveness results with other models.

#### 4.1.2. Movies Dataset

The Movies [21] dataset contains 200 clips extracted from action movies. Figure 11 shows an example of a clip classified as Fight (Fi).

The Movies dataset is a set of videos with characteristics similar to the Hockey dataset as they respond to a prepared and planned scene. However, it is more heterogeneous in terms of showing different scenes. Although the majority comprise boxing, there are many biases and it cannot be considered an adequate dataset for use on a violence recognition model in real video surveillance cameras.

#### 4.1.3. RWF-2000 Dataset

The RWF-2000 [22] dataset is the most significant violence detection dataset. It contains 2000 videos extracted from YouTube, each lasting 5 s. Figure 12 shows clips of videos labeled as violent.

This dataset has a greater diversity of videos, as it presents violent scenes from video surveillance cameras. However, the videos have previously been prepared, cut, corrected, and edited, as YouTube is the collection source. It also presents videos taken from smartphones and indoor scenes and contains almost no night scenes. These characteristics are not necessarily representative of video surveillance scenes; however, it can be used as a reference for comparing results with the state of the art.

#### 4.1.4. VioPeru Dataset

The various biases contained within the Hockey Fights [21], Movies [21], and RWF-2000 [22] datasets were extensively discussed. We concluded that they are not suitable for training a model oriented toward real-world scenarios.

Accordingly, as part of this research, we produced a new dataset called VioPeru, which consists of 280 videos collected from real video surveillance camera records. The videos were collected from the citizen security offices of the municipalities of Talavera, San Jeronimo, and Andahuaylas, in the Apurimac region, Peru.

Vioperu also includes 87 non-violent videos for false positive validation. Validation of false positives is crucial for better understanding our proposal’s limitations, especially to guarantee minimal human intervention in a real video surveillance system.

This compilation was compiled thanks to an agreements between the Jose Marea Arguedas National University and the three municipalities of the Apurimac region. Our team formally requested the videos; however, we did not participate in choosing the videos, much less in labeling the violent and non-violent videos. Finally, our team obtained the authorization documents for the use and publication of the videos for research purposes, which guarantee the dataset’s legitimacy and covers the ethical principles.

To guarantee freedom from bias, the original videos were taken in their entirety, respecting the number, resolution, time of day, and original sources. The selection and identification of violent videos was performed by the personnel of the citizen security offices (see Figure 13). Our editing work was limited to cutting the videos into 5 s clips.

Analyzing the videos provided made it possible to identify the following relevant characteristics:Violent scenes involve two people, several people, or crowds;Cameras have different resolutions;Violent scenes can be a very different size to the size of the frame; that is, the violent action can be so large or so small that it can go unnoticed by the human eye;Violent human actions occur primarily at night when lighting can negatively influence detection;Violence in video surveillance is not only made up of fights; looting, vandalism, violent protests, attacks on property, and confrontations between groups of people also occur;Occlusion is typical in video surveillance; that is, the people involved, trees, and vehicles, among others, can obscure violent scenes.

These characteristics varied and were determined to not cause biases in the dataset. Recently, these characteristics were identified as challenges [20] that have yet to be addressed in the field of the recognition of violent human action.

The VioPerú dataset served as the basis for the generation of our model, which was produced to recognize violent human actions for real-world scenarios, i.e., a model oriented toward video surveillance.

The dataset is available at https://github.com/hhuillcen/VioPeru (accessed on 1 January 2024).

### 4.2. Model Configuration

We used Python version 3.8 and PyTorch version 1.7.1 library as a base. The hardware was a workstation with NVidia GeForce RTX 3080 Ti GPU, 32 GB RAM, and a 32-core Intel Core i9 processor.

The datasets were divided into training and test subsets of 80% and 20%, respectively.

The following configuration was used in the training phase:Learning rate: $10^{-3}$ for all datasets;Batch size: 2;Number of epochs: 100;Optimizer: Adam, with Epsilon: $10^{-9}$; weight decay: $10^{-2}$; and Cross Entropy to calculate the loss function;One-Cycle Learning Rate Scheduler, with min-lr: $10^{-8}$; patience: 2; and factor: 0.5.

### 4.3. Evaluation of Results

#### 4.3.1. Evaluation of Results on Classical Datasets

The results were extracted from the *accuracy* metric in the classic datasets. Table 4 shows the results. A comparison with the most significant proposals is also shown.

The results show a comparison to other state-of-the-art proposals in terms of the effectiveness of the models. We observed cutting-edge results and contributions to the state of the art when performing the analysis for the Hockey dataset. Our proposal achieved 98.2% and was only surpassed by the 3D CNN end-to-end model [43] with 98.3%, with a 0.1% difference. However, it must be noted that this proposal is based on a 3D CNN, and given its high number of d and FLOPS parameters, it is not feasible for real-time use.

For the case of the Movies dataset, the results of our proposal reached the maximum accuracy, like the other proposals, i.e., it produced state-of-the-art results. It is necessary to indicate that our dataset has simple characteristics compared to the other datasets, such as RWF-2000.

#### 4.3.2. Evaluation of Results on the RWF-2000 Dataset

Unlike classical datasets, RWF-2000 [22] is the reference dataset for proposals aimed at effectiveness and efficiency. As a result, there are proposals using current techniques with better general results; thus, this is a good criterion for comparison.

As in the previous case, the *accuracy* metric was used to analyze the effectiveness of the models. For the efficiency results, the FLOPS and the number of parameters calculated by each model were used, and the results were compared with the most representative proposals from the state of the art.

Table 5 and Figure 14 show the results and a comparison in terms of effectiveness and efficiency.

It was observed that our proposal had the lowest amount of FLOPS, with a value of 3.15, after SepConvLSTM [47], with a value of 1.93. The other proposals had much higher values in this regard. This result suggests that, our model, when recognizing violent human actions, has a very short latency, which would allow for its use in devices with limited computational power. That is, the proposal can be used in real video surveillance scenarios. In addition, this result contributes to the state of the art regarding the efficiency of models for recognizing violent human actions.

As for the effectiveness results, in terms of *accuracy*, it was also observed that our proposal had a value of 88.5% and was only below SepConvLSTM [47] with a value of 89.75%. The other proposals had close results but were inferior to our proposal. This result demonstrates that our proposal has cutting-edge effectiveness, contributes to the state of the art, and can be used in real-world situations for violence identification in video surveillance cameras.

Finally, for the efficiency results in relation to the number of parameters, our proposal achieved better results than SA + TA [29], with a value of 5.29 million parameters, and both were below SepConvLSTM [47], with a value of 0.33 million parameters. However, when analyzing the complexity of the modules of our proposal, it was noted that the Short Temporal Extractor (STE) module uses the 2D CNN network MobileNetV2 [34], which has 3.4 million parameters; the rest of the modules only occupy 0.11 million parameters, i.e., the majority of the complexity of our proposal is the result of using the MobileNet V2 network [34].

The same analysis was performed for the SepConvLSTM proposal [47]. It was observed that it is composed of a two-stream architecture: a first stream with the background suppression technique, followed by a 2D CNN MobileNet V2 network [34], and then with separate convolutional LSTM layers; a second stream with frame difference, followed by another 2D CNN MobileNet V2 network [34], and then with separate convolutional LSTM layers, before the flows are finally joined with the final classifier. It is known that the number of parameters of MobileNet V2 is 3.4 million; when using this network for each stream, there are only 6.8 million parameters in the 2D CNN networks. In this way, it is not easy to understand how the entire model presented by SepConvLSTM [47] only has 0.33 million parameters.

According to our analysis, for a practical and real comparison of the efficiency results, we did not use the SepConvLSTM proposal [47] or the flow-gated network [22] because they use preprocessing based on optical flow, increasing the computational cost of the models to levels that are unfeasible for real-time applications. Therefore, after discarding these two proposals, our proposal was the best in terms of efficiency, with 3.51 million parameters and 3.15 GFlOPs, and it remained the second best in terms of effectiveness, reaching an accuracy of 88.5%.

#### 4.3.3. Results Evaluation on the VioPeru Dataset

Since the SepConvLSTM proposal [47] remained superior to our proposal in terms of effectiveness, i.e., *accuracy*, it was a good candidate for testing with the dataset presented in this research: VioPeru. In this way, the *accuracy* of both proposals was calculated.

The VioPeru dataset was divided into samples consisting of 80% for training and 20% for testing. The configuration of our model was the same as detailed in the model configuration subsection.

In the case of SepConvLSTM [47], the same characteristics described in the scientific article were considered. Only 32 frames were taken from each video using uniform sampling, and frames were resized to 320 × 320. Before use with the model, they were cropped to random sizes and resized to 224 × 224. Data augmentation techniques, such as random brightness, random cropping, Gaussian blur, blurs, and horizontal modifications, were used. The training was performed for approximately 150 epochs. The CNNs were initialized using weights pretrained on the ImageNet dataset, and Xavier initialization was used for the SepConvLSTM kernel. For the optimization model, the AMSGrad variant of the Adam optimizer was used. Training began with a learning rate of 0.0004. After every five epochs, the learning rate was reduced by half until it reached 0.00005. Table 6 shows the results of both proposals.

It was observed that, in the case of the VioPeru dataset, our proposal was far superior to SepConvLSTMN [47] in terms of effectiveness, reaching an *accuracy* of 89.29% compared to 73.21% for SepConvLSTMN.

This result should not be analyzed in numbers alone, although our proposal was superior. However, it is necessary to clarify that the VioPeru dataset consists of scenes of real violence and non-violence extracted from real video surveillance cameras in the province of Andahuaylas in Peru. Therefore, it was used as a reference dataset to test our proposal from the point of view of its validation in a real-time scenario, which was ultimately the objective of our research.

This result also demonstrates that our model was not aimed at producing results using datasets with mixed and varied videos, but rather at becoming a general-use model oriented towards the domain of violence detection in video surveillance cameras in real time, i.e., it was not only efficient, but it was also effective in real-world scenarios and on state-of-the-art datasets.

#### 4.3.4. Evaluation of Results in Real Time

To the best of our knowledge, there is no formal method to measure whether a model can be used in real time. However, based on similar work [29,44], we decided that evaluating results in real time should be performed by measuring the processing time for every 30 frames, assuming that video surveillance videos have this default setting, i.e., a speed of 30 frames per second.

For this analysis, a prototype of a local video surveillance system was utilized (see the next sub-subsection) to carry out operational tests in real time.

The processing time of our proposal was measured for every 30 frames. For this, a laptop with a 2.7 GHz 13-core Intel Core i7 processor, 16 GB RAM, and an NVIDIA Quadro P620 graphics card with 2 GB GPU memory was used.

The result was 0.0720 s on average, which is the latency time of the model when processing 30 frames. In other words, our proposal only needs 0.0720 s to process a 1 s video. If we consider that real time has a latency of 0 s, the result is very close.

This evaluation should not only be interpreted in numbers but also in a practical-use sense: our model can be used in any video surveillance system. By simply connecting the video inputs of the cameras to one or more devices with our model deployed, the model will recognize violent scenes in real time, alerting staff and following the corresponding security protocol.

These tests confirmed that the proposal is lightweight and works on devices with low computational power, with cutting-edge results in terms of effectively detecting violence in real time.

#### 4.3.5. Implementation of a Local Video Surveillance System Prototype

The introduction and identified problems sections note that the current proposals are not oriented towards real video surveillance scenarios. Our work addresses this challenge. Despite our proposal not being deployed in a real video surveillance system, we implemented it on a local video surveillance system prototype for devices with low computational power. The idea was to test our proposal in simulated situations of violence and non-violence and observe its behavior. In addition, we wanted to evaluate the latency of recognition in a real-world scenario of violence and non-violence in such a way as to produce indicators for a real-time evaluation.

The latency of the prototype depended on the computing power of the equipment. We used a laptop with a 2.7 GHz 12-core Intel Core i7 processor, 16 GB RAM, and an NVIDIA Quadro P620 graphics card with 2 GB GPU memory.

The deployed prototype ran in an infinite loop displaying video frames from the camera at a normal speed of 30 frames per second. Each loop captured 30 frames and stored them in a buffer; the frames were resized to 240 × 240 pixels and then analyzed with our previously generated model. The model outputs were a violence or nonviolence label, which was displayed as a text box over the video frames, before the buffer was finally released. The same was executed for the next 30 frames in the next loop, and so on (see Figure 15).

## 5. Ablation Study

### 5.1. Analysis of Results on Combinations of the Proposed Modules

To determine the degree of contribution of the three modules to the proposal, a study was carried out to analyze the behavior of the modules in terms of effectiveness and efficiency and their contribution to the proposal.

To execute this, the modules were tested independently on the RWF 2000 dataset [22], with some combinations of techniques in the Spatial Motion Extractor (SME) module (see Table 7).

The Spatial Motion Extractor (SME) module did not produce results by itself since it only extracts regions of interest. Therefore, it was combined with the Short Temporal Extractor (STE) module for the analysis. However, a combination of techniques was used to extract regions of interest: a technique that implements two AVG pooling layers, followed by four convolutional layers with ReLU. The other technique used image processing functions, specifically morphological deformations.

The table shows that the module: SME with (2 AVG Pool and 4 Conv/Relu) + STE, achieved an acceptable accuracy result of 82.5%. The number of parameters and FLOPS were within expectations, but they can be considered contributions.

The SME module with (Morphological Deformations) + STE substantially improved the model’s effectiveness, increasing the accuracy by 2.75%. A slight decrease was also observed in the number of parameters and FLOPS. This combination was efficient but did not produce the best effectiveness results. This is because the STE module only recognizes quick and short movements, and it still fails in longer movements.

Finally, the Global Temporal Extractor (GTE) module was added, significantly improving the efficiency, which reached an accuracy of 88.5%, with a slight increase in the number of parameters and FLOPS.

### 5.2. Analysis of the Results Using Backbone Combinations

According to the proposal presented in Section 3, the Short Temporal Extractor (STE) module uses a pretrained 2D CNN network as a backbone. The proposal uses the MobileNet V2 architecture [34], mainly due to the cutting-edge results obtained using ImageNet [40] with fewer parameters and FLOPS. However, according to Table 1, there are other good candidate backbones, such as EfficientNet B0 [36], MobileNet V3 L [35], and MnasNet [37].

This section evaluates the proposal with the above-mentioned models regarding effectiveness and efficiency on the RWF 2000 [22] and VioPeru datasets. Table 8 shows the results.

Using the VioPeru dataset, our proposal had the best efficiency with the MobileNet V2 backbone, with an *accuracy* of 89.29%, followed by MobileNet V3 [35] with 89% and EfficientNet B0 [36] with 87.5%. In addition, our proposal had a better efficiency with the MobileNet V2 backbone [34], with a value of 3.51 million parameters and 3.15 GFLOPS.

With the RWF-2000 dataset [22], our proposal also had the best efficiency with the MobileNet V2 backbone, with an *accuracy* of 85.5%, followed by MobileNet V3 [35] and EfficientNet B0 [36] with the same value of 88.25%. Although the proposal with the MNasNet [37] backbone had the best efficiency, with a value of 2.22 million parameters and 1.13 GFLOPS, it had very low effectiveness compared to the other backbones.

Thus, we chose the MobileNet V2 backbone as a result of the best results in terms of effectiveness and efficiency.

### 5.3. Quality Analysis

This section analyzes the quality of the results of our proposal on the VioPeru dataset. Six misclassified videos were extracted: two non-violent and four violent. Figure 16 shows the six videos as a sequence of six frames each, at an interval of 1 s.

The first row of Figure 16 corresponds to the frames of the non-violent video “nf53sanj.avi”. The video is low resolution and shows two girls playing with their hands in the distance. The rapid hand movement, low resolution, and remoteness of the violent scene confused our model, which classified it as violent.

The second row corresponds to the frames of the non-violent video “nf99and.avi”. A person is observed insistently knocking on a door with his hand. Like the previous case, this rapid movement confused the model, which classified it as violent.

The third row corresponds to the frames of the violent video “f80and.avi”. A crowd of people is observed; the scene is ambiguous, as there is movement of people preparing for a violent confrontation with the police, but no rapid movements are observed. These factors confounded the model, which classified it as non-violent.

The fourth row corresponds to the violent video “f67and.avi”. A crowd is observed at the end of a street fight which is occluded with artificial light, and the scene is unclear. The violence has mostly finished, but people are running. These factors confused the model, which classified it as non-violent.

The fifth row corresponds to the frames of the violent video “f96and.avi”. The video shows a crowd of civilians and police; everyone is running, people are blocked from each other, smoke and fire are observed, and everything is very hard to make out. The model classified it as non-violent, as the action of running was not recognized as violence by the model.

Finally, the sixth row corresponds to the video “f137and.avi”. This shows a distant scene with a crowd committing acts of vandalism. There are no quick movements except running, which confused the model, making it recognize it as non-violent.

According to this analysis, the limitations of our proposal are as follows:The model needs to be more robust when analyzing videos with certain rapid movements, especially with people’s hands or arms, which confuse recognition;The model is confused when scenes show certain types of crowds with people running and some occlusion.

However, it is necessary to clarify that the proposed model correctly classifies most of the situations and scenes described above. These limitations and weaknesses were only observed in some scenes.

### 5.4. Individual Analysis of Positive and Negative Samples

VioPeru has 280 videos: 140 violent and 140 non-violent; for the analysis of our model, the dataset was divided into an 80% proportion for training and a 20% proportion for testing, making the number of test videos in this dataset only 56: 28 violent and 28 non-violent.

It is necessary to do an individual analysis of the behavior of our model in the violent and non-violent videos of the entire VioPeru dataset. For this, all the videos were processed, and the results are shown in Table 9.

Our model performs better at detecting violent videos than non-violent videos, although by a small margin. It is necessary to clarify that for the collection of violent and non-violent videos from VioPeru, a violent scene was previously identified. Subsequently, the violent clips were extracted, and the non-violent videos were extracted from the non-violent part prior to the identified scene. The objective was for the model to learn and identify the limit of what is violent and non-violent. For this reason, the model could have confusion when identifying false positives from VioPeru.

### 5.5. Analysis of False Positives in Non-Violent Videos

We consider that violence is an atypical activity in a real video surveillance system concerning activities that are not violent. Therefore, it is essential to analyze the resistance of our proposal to false positives in order to provide results that lead to a better understanding of the minimum human participation in a real video surveillance system.

To this end, the number of non-violent VioPeru test videos is insufficient for this analysis. A compilation of 87 new non-violent videos was made from different surveillance cameras of the security systems of three districts of the Andahuaylas in Peru. They are videos of different city streets, with normal human activities, daytime and nighttime videos, and with and without crowds. Figure 17 shows random frames from some of these videos.

The videos are available in the “false_positives_validation” folder within the published VioPeru dataset: https://github.com/hhuillcen/VioPeru (accessed on 1 January 2024).

By subjecting these videos to our model, an accuracy of 98.85% was reached; that is, only one false positive was identified, and it corresponds to the video “val_nfsanj46.avi”, the Figure 18 some video frames:

The real scene corresponds to two people talking, and one falls; the other person tries to contain him but fails. It is a confusing video even for a human being, as it is confused as a violent activity of intentionally causing the fall. This same confusion caused the model to recognize the scene as violent.

In general, there are some rare scenes that can confuse any model, including human recognition. However, our model recognized the other videos correctly; we can affirm that our proposal provides results that enable minimal human participation in the violence detection process in real video surveillance systems.

## 6. Conclusions

In this work, a model based on deep learning for the recognition of violent human actions in real-time video surveillance is proposed. We propose an architecture with three modules. The first module, the Spatial Motion Extractor (SME), extracts regions of interest from a frame using frame difference and morphological dilation. The second module, the Short Temporal Extractor (STE), extracts temporal features from fast and short-duration movements through temporal fusion and the use of the MobileNet V2 backbone. Finally, the Global Temporal Extractor (GTE) module identifies long-term temporal characteristics and fine-tunes the model for better precision, using global average pooling and dot product. Tests were initially carried out on the RWF-2000, Movies, and Hockey datasets, producing cutting-edge results in terms of both effectiveness and efficiency. In order to demonstrate that the aforementioned datasets are not oriented to video surveillance, a dataset called VioPeru was generated with real videos from video surveillance cameras in Peru. The results show that our proposal is the best in terms of efficiency and effectiveness on VioPeru. The proposal exhibited a recognition latency of 0.0720 s for every 30 frames, which is close to real time. Our proposal exhibited high efficiency and effectiveness in real-time video surveillance systems and can be used in devices with low computational power.

## 7. Future Work

After testing our proposal on the VioPeru dataset and achieving an efficiency of 88.5%, with SepConvLSTM only reaching 73.21%, the challenge remains to surpass these results in terms of accuracy.

VioPeru is a compilation of videos of violence obtained from real video surveillance cameras. The violent scenes are extremely varied and produce various challenges, e.g., crowd situations, day and night scenes, occlusion, heavy artificial light, high and low resolutions, and different types of violence. Our proposal was tested using VioPerú without distinguishing between these particularities; however, other proposals test their results according to performance for each independent characteristic. This is because certain models have certain biases in different situations.

One of the critical factors in achieving the high effectiveness and efficiency when testing our model is the Spatial Motion Extractor (SME) module, which behaves as an attention mechanism. Future proposals should to recognize attention as a critical factor; in fact, the transformer technique uses it. However, it has a high computational cost and is inefficient for real-time applications. In this way, we hope to influence future techniques in improving care mechanisms from the point of view of efficiency.

Our study was not tested using a real video surveillance system but by measuring latency times on devices with low computing power. In future work, we hope to test new models for real video surveillance systems, e.g., by describing the hardware scenario, in order to obtain better effectiveness, efficiency, and real-time accuracy results.

A weakness detected in our proposal is the confusion in specific fast movements. The short temporal extractor (STE) module can be improved by taking non-contiguous consecutive frames, for example, consecutive frames with intervals of three or four frames. This modification could make the difference between rapid movements clearer in identifying violence. In addition, it will make the model lighter.

## Figures and Tables

**Figure 1 sensors-24-00668-f001:**
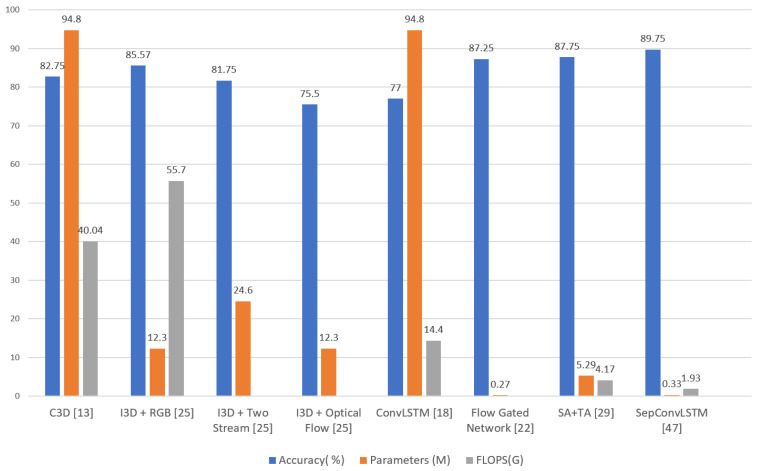
Graphic Comparison of of the methods for recognizing violent human actions in video surveillance using the RWF-2000 dataset.

**Figure 2 sensors-24-00668-f002:**
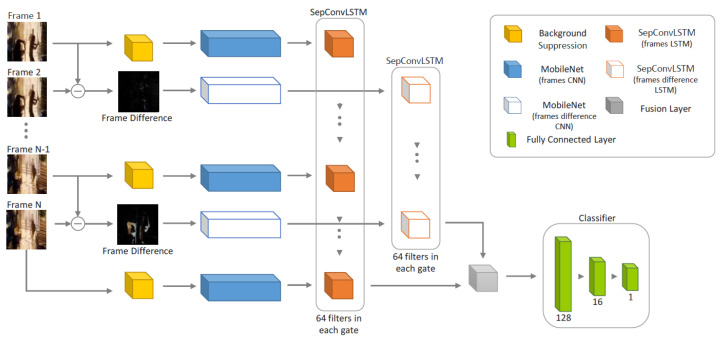
The architecture of the proposal presented by Islam et al. [47].

**Figure 3 sensors-24-00668-f003:**
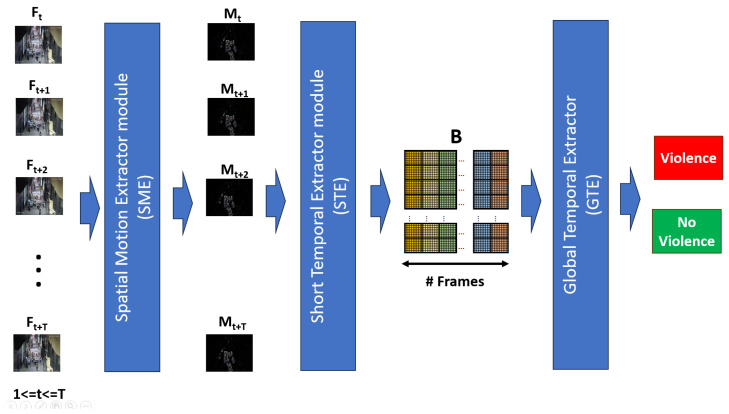
Summary of the proposed architecture.

**Figure 4 sensors-24-00668-f004:**
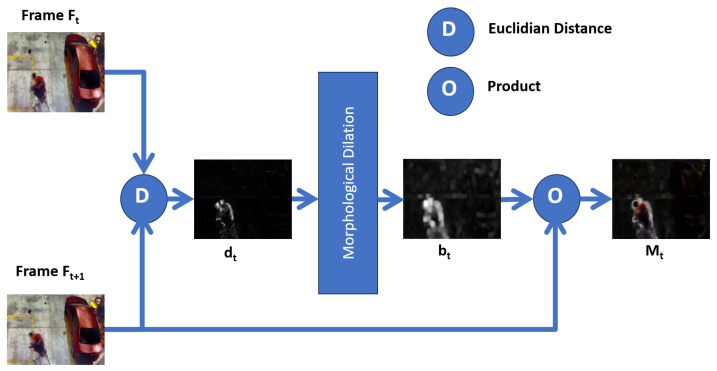
Spatial Motion Extractor (EME) module.

**Figure 5 sensors-24-00668-f005:**
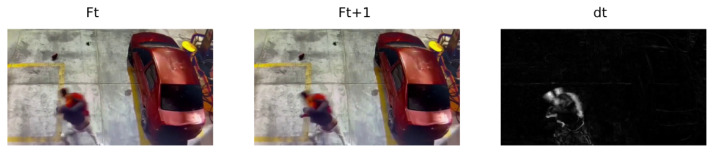
Euclidean distance between two consecutive frames $F_{t},F_{t+1}$.

**Figure 6 sensors-24-00668-f006:**
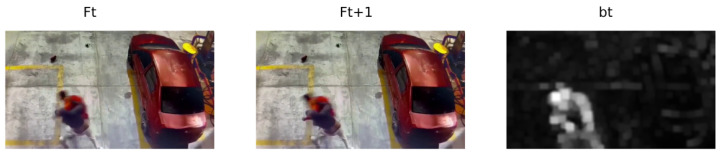
Morphological deformations in $d_{t}$.

**Figure 7 sensors-24-00668-f007:**
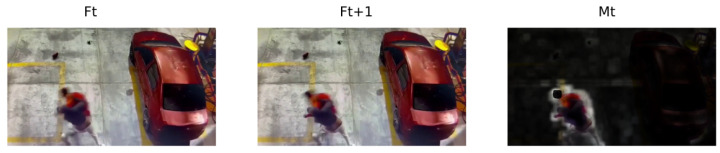
$M_{t}$: spatial extraction of motion.

**Figure 8 sensors-24-00668-f008:**
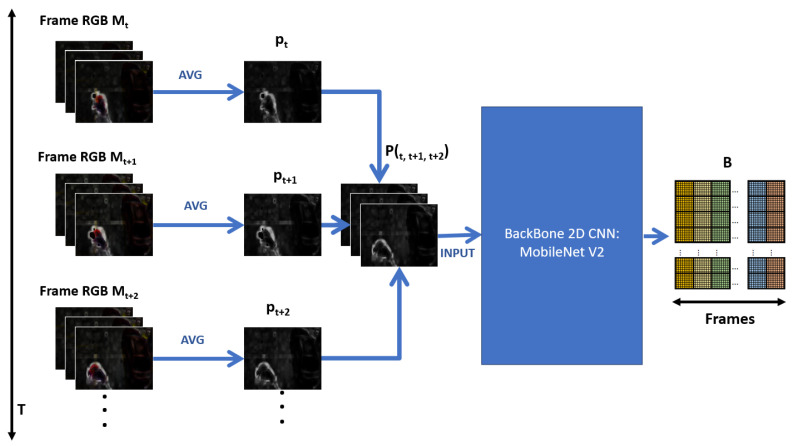
Short Temporary Extractor (STE) module.

**Figure 9 sensors-24-00668-f009:**
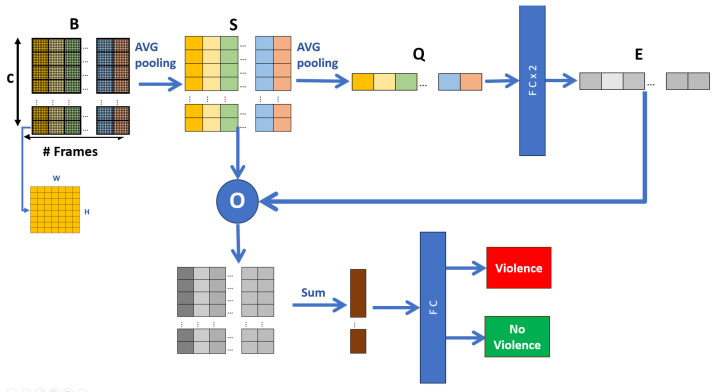
Global Temporal Extractor (GTE) module.

**Figure 10 sensors-24-00668-f010:**
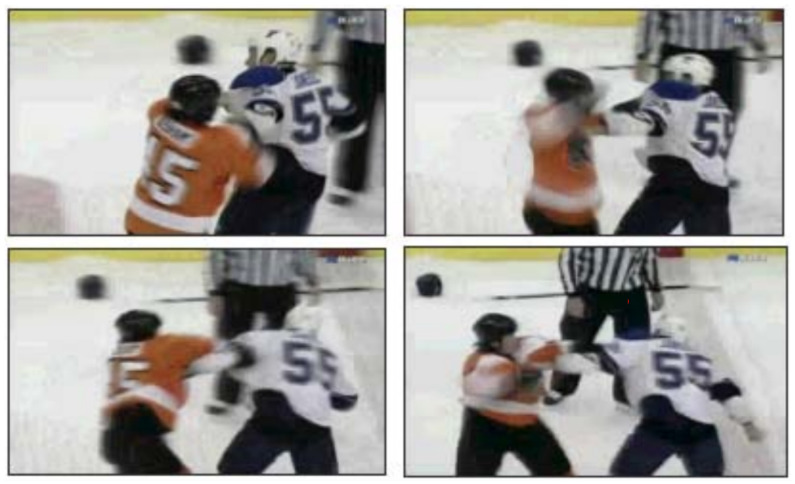
Frame sequence of a video from the Hockey Fights dataset.

**Figure 11 sensors-24-00668-f011:**
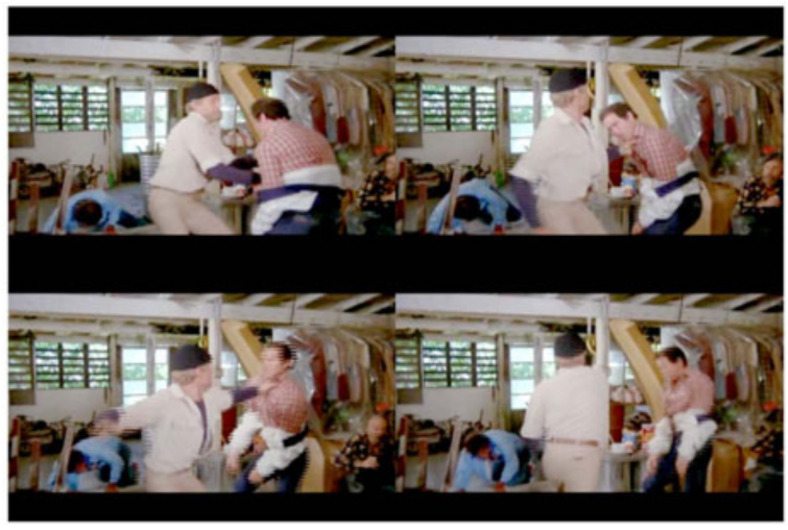
Frame sequence of a video from the Movies dataset.

**Figure 12 sensors-24-00668-f012:**
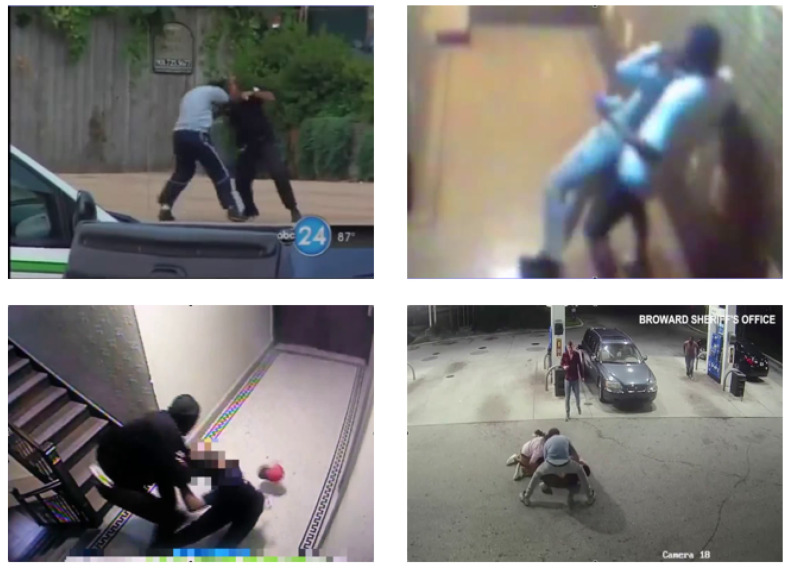
Examples of violent clips from the RWF-2000 dataset.

**Figure 13 sensors-24-00668-f013:**
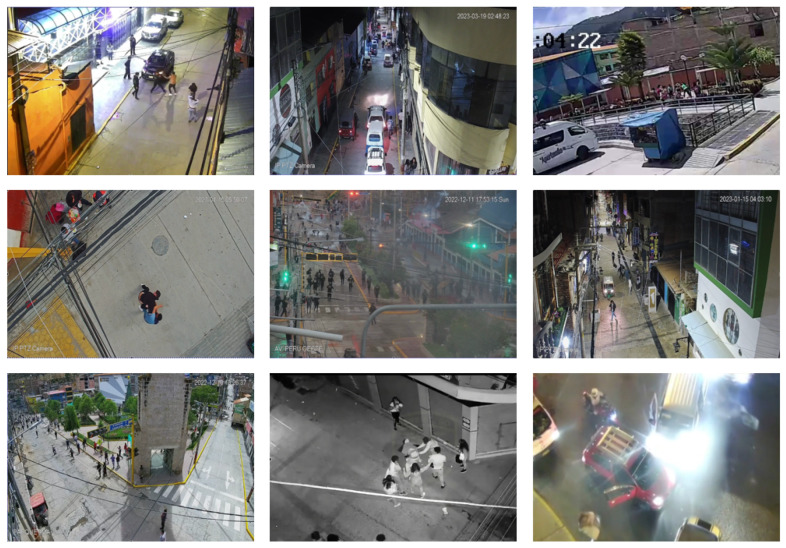
Examples of violent frames from the VioPeru dataset.

**Figure 14 sensors-24-00668-f014:**
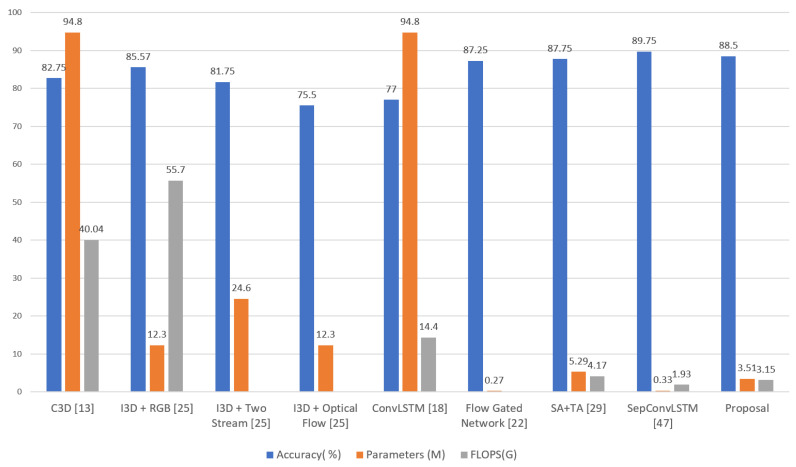
Graphic comparison of results obtained in the RWF-2000 dataset, taking efficiency and effectiveness as a reference.

**Figure 15 sensors-24-00668-f015:**
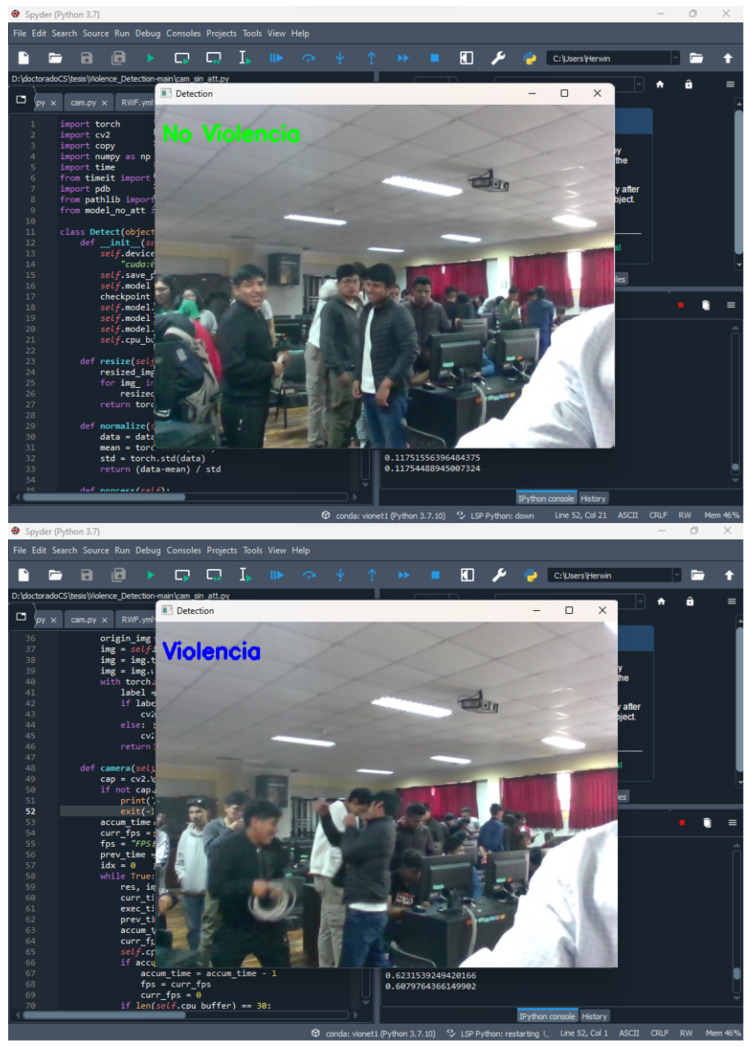
Deployment of the video surveillance system prototype.

**Figure 16 sensors-24-00668-f016:**
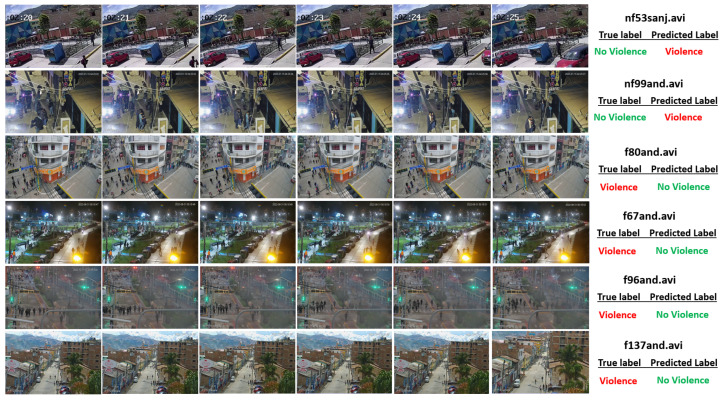
Frame sequence of six misclassified VioPeru videos.

**Figure 17 sensors-24-00668-f017:**
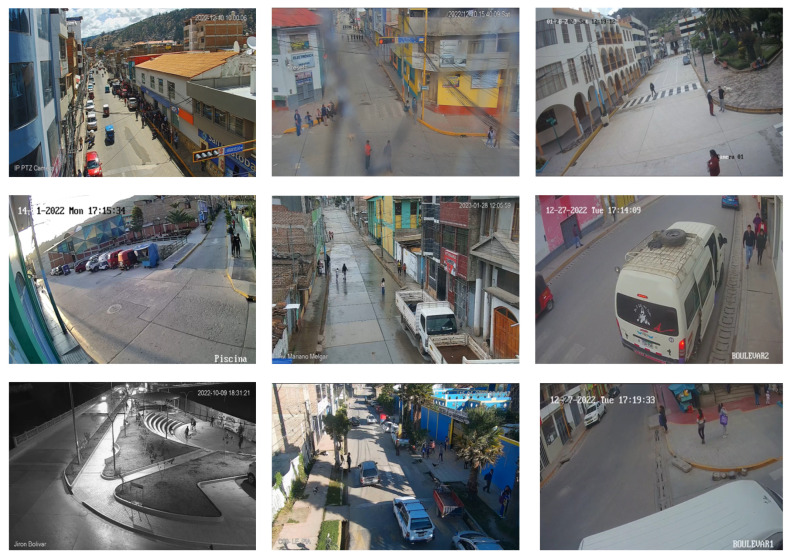
Random videos of 87 non-violent videos for false positive analysis.

**Figure 18 sensors-24-00668-f018:**
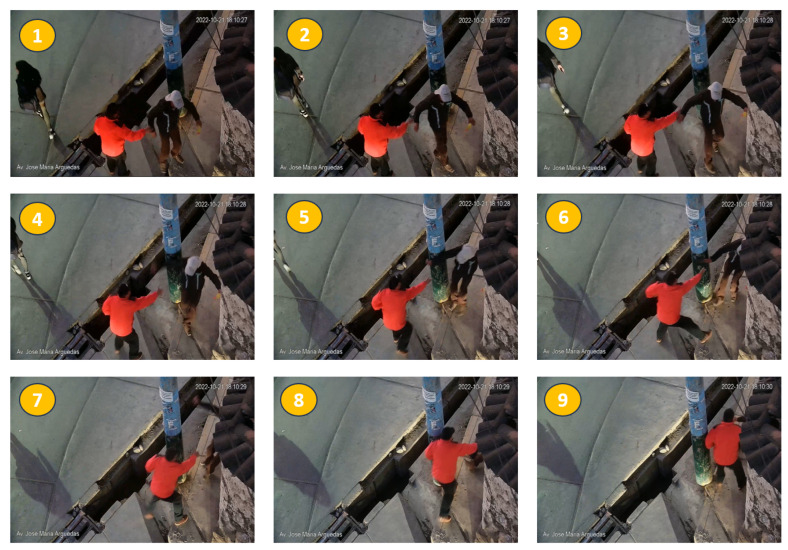
Frame sequence of the video recognized as false positive. The image shows a sequence of nine frames of a nonviolent scene; one person falls, and the other tries to hold them up; the movements are confusing and cause the model to classify it as violent.

**Table 1 sensors-24-00668-t001:** Summary of effectiveness and efficiency results of backbone models tested on the ImageNet dataset [40].

Model	Accuracy (%)	Number of Parameters (M)	FLOPS (G)
Resnet50	76.0	25.6	3.8
InceptionV3	78.8	23.2	5.0
DenseNet121	74.0	8.0	2.8
Squeezenet	57.5	1.25	0.83
GhostNetV2	75.3	12.3	0.39
EffcientNetB0	78.0	5.3	1.8
MobileNetV2	72.6	3.4	0.3
MobileNetV3 L	76.6	7.5	0.36
MasNet	75.2	3.9	0.315
Vision Transformers (VIT-Huge)	88.55	632.0	-

**Table 2 sensors-24-00668-t002:** Summary of the methods for recognizing violent human actions in video surveillance using classic datasets.

Method	Hockey Fight Dataset	Movies Dataset	Violent Flow Dataset
ViF + OViF [6]	87.5 ± 1.7%	-	88 ± 2.45%
Radon Transform [7]	90.1 ± 0%	98.9 ± 0.22%	-
STIFV [8]	93.4%	99%	96.4%
MoIWLD [9]	96.8 ± 1.04%	-	93.19 ± 0.12%
OR-VLAD [10]	98.2 ± 0.76%	100 ± 0%	93.09 ± 1.14%
Three streams + LSTM [15]	93.9%	-	-
FightNet [16]	97.0%	100%	-
Hough Forests + CNN [7]	94.6 ± 0.6%	99 ± 0.5%	-
ConvLSTM [18]	97.1 ± 0.55%	100 ± 0%	94.57 ± 2.34%
Bi-ConvLSTM [19]	98.1 ± 0.58%	100 ± 0%	93.87 ± 2.58%
3D CNN end to end [43]	98.3 ± 0.81%	100 ± 0%	97.17 ± 0.95%
3D-DenseNet (2,6,12,8) [44]	97.0%	100%	90%
SA+TA [29]	97.2%	100%	-

**Table 3 sensors-24-00668-t003:** Summary of the methods for recognizing violent human actions in video surveillance using the RWF-2000 dataset.

Model	Accuracy (%)	Parameters (M)	FLOPs (G)
C3D (Tran et al.) [13]	82.75	94.8	40.04
I3D + RGB (Carreira et al.) [25]	85.57	12.3	55.7
I3D + Two Stream (Carreira et al.) [25]	81.75	24.6	-
I3D + Optical Flow (Carreira et al.) [25]	75.5	12.3	-
ConvLSTM (Sudhakaran et al.) [18]	77.0	94.8	14.4
Flow Gated Network (Cheng et al.) [22]	87.25	0.27	-
SA + TA (Huillcen et al.) [29]	87.75	5.29	4.17
SepConvLSTM (Islam et al.) [47]	89.75	0.33	1.93

**Table 4 sensors-24-00668-t004:** Comparison of results obtained on classical dataset.

Method	Hockey Fight Dataset	Movies Dataset	Violent Flow Dataset
ViF + OViF [6]	87.5 ± 1.7%	-	88 ± 2.45%
Radon Transform [7]	90.1 ± 0%	98.9 ± 0.22%	-
STIFV [8]	93.4%	99%	96.4%
MoIWLD [9]	96.8 ± 1.04%	-	93.19 ± 0.12%
OR-VLAD [10]	98.2 ± 0.76%	100 ± 0%	93.09 ± 1.14%
Three streams + LSTM [15]	93.9%	-	-
FightNet [16]	97.0%	100%	-
Hough Forests + CNN [7]	94.6 ± 0.6%	99 ± 0.5%	-
ConvLSTM [18]	97.1 ± 0.55%	100 ± 0%	94.57 ± 2.34%
Bi-ConvLSTM [19]	98.1 ± 0.58%	100 ± 0%	93.87 ± 2.58%
3D CNN end to end [43]	98.3 ± 0.81%	100 ± 0%	97.17 ± 0.95%
3D-DenseNet (2,6,12,8) [44]	97.0%	100%	90%
SA + TA [29]	97.2%	100%	-
Proposal	98.2%	100%	-

**Table 5 sensors-24-00668-t005:** Comparison of results obtained in the RWF-2000 dataset, taking efficiency and effectiveness as a reference.

Model	Accuracy (%)	Parameters (M)	FLOPs (G)
C3D (Tran et al.) [13]	82.75	94.8	40.04
I3D + RGB (Carreira et al.) [25]	85.57	12.3	55.7
I3D + Two Stream (Carreira et al.) [25]	81.75	24.6	-
I3D + Optical Flow (Carreira et al.) [25]	75.5	12.3	-
ConvLSTM (Sudhakaran et al.) [18]	77.0	94.8	14.4
Flow Gated Network (Cheng et al.) [22]	87.25	0.27	-
SA+TA (Huillcen et al.) [29]	87.75	5.29	4.17
SepConvLSTM (Islam et al.) [47]	89.75	0.33	1.93
Proposal	88.5	3.51	3.15

**Table 6 sensors-24-00668-t006:** Comparison of results on the VioPeru dataset, taking efficacy as a reference.

Model	Accuracy (%)
SepConvLSTM (Islam et al.) [47]	73.21
Proposal	89.29

**Table 7 sensors-24-00668-t007:** Performance analysis of module combinations on the RWF 2000 dataset [22].

Module	Accuracy (%)	Parameters (M)	FLOPs (G)
SME with [2 AVG Pool and 4 Conv/Relu] + STE	82.5	3.49	3.15
SME with [Def. Morf.] + STE	85.25	3.47	3.13
SME with [Morf. Def.] + STE + GTE	88.5	3.51	3.15

**Table 8 sensors-24-00668-t008:** Proposal results with different backbones.

Proposal Variations	RWF-2000 Accuracy(%)	VioPeru Accuracy (%)	Parameters (M)	FLOPS (G)
Proposal with EfficientNet B0 backbone	88.25	87.5	5.29	4.17
Proposal with MobileNet V2 backbone	88.5	89.29	3.51	3.15
Proposal with MobileNet V3 backbone	88.25	89	7.62	4.1
Proposal with MNasNet backbone	75.25	62.5	2.22	1.13

**Table 9 sensors-24-00668-t009:** Individual analysis of positive and negative samples.

Sample	Videos	Accuracy
Violent	140	90.71%
Non-violent	140	89.29%

## Data Availability

The datasets VioPeru used in this work can be accessed on the following link: https://github.com/hhuillcen/VioPeru (accessed on 1 January 2024).

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
