# Peer review of "Efficient Human Violence Recognition for Surveillance in Real Time"

_sensors, 2024, doi:10.3390/s24020668_

Round 1

Reviewer 1 Report

Comments and Suggestions for Authors

1. The paper should introduce a more precise term for "generate a dataset" as it appears that the authors did not strictly "generate" a dataset in their study. Comprehensive details on the collection process and the data gathered are essential. The authors should explicitly ensure or provide information demonstrating that the dataset is free from bias. Additionally, insights into the privacy aspects and the usage of the dataset should be included.

2. The authors incorporated several modules into their study without clearly articulating the rationale for their selection over existing alternatives. It is noteworthy that the authors did not extensively showcase or enhance these modules, which leaves room for improvement.

3. A more comprehensive ablation study is recommended. With multiple modules, the authors should conduct experiments on specific combinations to precisely gauge the significance each module contributes to the overall performance. Practicality considerations of each module in achieving the model's objectives should also be explored.

4. The usage of commas instead of decimal points for percentages needs clarification from the authors.

5. The study lacks a sufficiently broad range of tests and metrics prior to endorsing the proposed model. It is advisable to compare the performance of the proposed work against other state-of-the-art models addressing the same problem.

6. The given equations, while standard, do not explicitly elucidate how the model achieves its intended performance. A more detailed explanation of the model's workings is required.

7. The reviewed literature in the domain appears limited. The authors should conduct a more extensive literature search to enhance the background of their research.

8. The reference to backbones lacks significance in the context of the proposed study. The overall discussion is insufficient, omitting crucial details and leaving a significant gap in understanding.

9. The rationale for proposing the solution does not align precisely with the current problem in the research domain. The authors should provide a more rigorous and scientific justification for their proposed solution, rather than assembling modules without a clear guiding principle.

10. The authors did not offer sufficient details or a mathematical explanation regarding why spatiotemporal aspects are crucial to the success of their study.

11. The study relies on existing modules combined and tested on their dataset without thoroughly addressing how such a solution can advance the principles of the domain. The evaluation and experiments conducted are subpar.

12. The conclusions should provide more guidance on how other researchers can build upon or improve the work, outlining a trajectory for future research.

13. Limitations of the study should be thoroughly discussed, supported by actual results. The authors should present instances where their model fails and provide explanations for such occurrences.

14. In comparison to other results, the authors' work exhibits only minimal improvement but a significant increase in cost. The authors should elaborate on why their solution is considered preferable despite this.

15. The VioPeru dataset is heavily compressed. The authors need to clarify how their study performs with real-world, real-time camera inputs. Providing a test dataset from a more realistic perspective and evaluating the model's performance on actions not considered in the dataset is crucial, especially as the authors position it as a general-purpose model for human violence.

16. The overall paper requires extensive proofreading and revision to enhance clarity and coherence.

Comments on the Quality of English Language

The authors should consider looking into and addressing my comments critically.

Author Response

Dear reviewer.
Thank you for your time and comments. I am sending a file with the responses to your comments.

Best regards.

Herwin Alayn Huillcen Baca

Reviewer 2 Report

Comments and Suggestions for Authors

The manuscript introduces an efficient deep learning model for real-time recognition of violent human actions in video surveillance. The proposed architecture, encompassing the Spatial Motion Extractor (SME), Short Temporal Extractor (STE), and Global Temporal Extractor (GTE) modules, is innovative and demonstrates a thoughtful fusion of spatial and temporal features. The creation of the VioPeru dataset is a significant contribution, addressing the limitations of existing datasets, and the reported outcomes in efficiency, effectiveness, and real-time applicability are promising. However, the manuscript requires major revisions, particularly in terms of language, structure, and addressing crucial aspects such as precision, and latency discussion.

Strengths:

1. Innovative Model Architecture: The integration of SME, STE, and GTE modules showcases an inventive approach to violence recognition, providing a strong foundation for the proposed model's effectiveness.

2. Dataset Contribution: The introduction of the VioPeru dataset is a good contribution, emphasizing the model's applicability in real-world surveillance scenarios.

3. Efficiency and Effectiveness: The manuscript effectively highlights the model's commendable performance, especially in terms of efficiency and effectiveness, indicating its potential for real-time applications.

Comments/Questions:

- Latency Discussion: A more detailed discussion on the achieved latency and its practical implications in real-time applications is necessary for a comprehensive understanding of the model's capabilities.

- Handling Multiple Persons: Given the importance of surveillance scenarios involving more than two persons, the authors are encouraged to discuss how their model addresses or could be extended to handle such situations.

- Precision and Recall Reporting: The manuscript lacks a comprehensive report on precision and recall metrics for the classification model, which is essential for a holistic evaluation of the proposed method.

- Assumption Clarification: The assumption of rapid movement for violent actions should be clarified, acknowledging that the method is specifically tailored for rapid violent actions.

- Redundancy in Abstract: The redundant sentence in the Abstract should be revised for clarity and conciseness, e.g., the last sentence.

- Language and Structure Revisions:

·       Clarity and Coherence: The language used in the manuscript lacks clarity, making it challenging for readers to follow the logic and understand the proposed concepts. Sentences and paragraphs should be constructed with clarity in mind, ensuring that each point logically follows the previous one.

·       Paragraph Usage: The manuscript exhibits issues related to paragraph usage. Paragraphs should be appropriately used to group related ideas or concepts. A new paragraph should be started when there is a shift in topic or when introducing a new point.

·       Punctuation Errors: There are instances of incorrect punctuation, such as the misuse of commas, periods, or other punctuation marks. Proper punctuation is essential for conveying ideas accurately and maintaining readability.

·       Grammar: The review suggests that there are grammatical errors in the manuscript. This includes issues such as subject-verb agreement, or incorrect word usage.

- Figure Font Size and Vector Graphics: Figures should adhere to appropriate font sizes, and the use of vector graphics is recommended for improved quality.

- Cross-Reference Error: The cross-reference error at the end of line 452 needs correction for accuracy.

- Real-Time Performance Clarification: The paragraph discussing real-time performance (lines 554-558) requires clarification and correction of reported latency times.

In conclusion, while the manuscript demonstrates a promising contribution, addressing the outlined major revisions is crucial for enhancing clarity, precision reporting, and overall quality, making it suitable for publication.

Comments on the Quality of English Language

The quality of English should be improved to enhance the paper's presentation.

Author Response

(The authors gave the same response as above.)

Reviewer 3 Report

Comments and Suggestions for Authors

In the paper “Efficient Human Violence Recognition for Surveillance in Real-Time” the authors propose a new real-time violence detection model which is designed for real-life scenarios. To evaluate the approach, 4 different datasets were used. The structure is good, the key ideas of proposed method are clear. The authors also proposed the new dataset VioPeru, which is a good contribution to automatic video surveillance area. 

The authors should provide more details about training process in section 4.3.3, especially about SepConvLSTM model. Also, since the authors claimed that they designed  a method for real-life scenarios, its evaluation on a few long additional non-violent videos would be a great addition to the article. Violence is an extremely rare occurrence, and for real systems it is also important to have a minimum number of false positives. However, many academic datasets (including the proposed dataset) do not have large enough samples and are 50/50 balanced. Because of this, it is difficult to understand whether the proposed approach will work correctly with minimal human involvement to check the result of the system.

The work also contains many typos (lines 220-222, 452, 513, 525, 555) and requires careful reading.

Overall, the article cannot be accepted in its current form and needs improvement. From the scientific side, it does not bring any significant results. On the engineering side, there are doubts about the correct operation of the system in real conditions - additional experiments are needed to determine the reliability of the system. o determine the reliability of the system.

Comments on the Quality of English Language

Minor editing of English language required

Author Response

(The authors gave the same response as above.)

Round 2

Reviewer 2 Report

Comments and Suggestions for Authors

Author Response

Dear reviewer

Thank you very much for taking the time to review this manuscript. We see that you no longer have comments.

Best regards

Herwin Alayn Huillcen Baca

Reviewer 3 Report

Comments and Suggestions for Authors

In the paper “Efficient Human Violence Recognition for Surveillance in Real-Time” the authors propose a new real-time violence detection model which is designed for real-life scenarios. To evaluate the approach, 4 different datasets were used. The structure is good, the key ideas of the proposed method are clear. The authors also proposed the new dataset VioPeru, which I downloaded and checked for similarities between the train and validation clips, and everything looks fine. Therefore, the proposed dataset is a good contribution to the automatic video surveillance area.

The main doubt about the proposed system is its resistance to false positives. Violence is an extremely rare occurrence, and for real systems it is also important to have a minimum number of false positives.  The authors claimed that they designed it for real-life scenarios, so it needed to be evaluated on several long additional videos without violence. This could be some YouTube videos with street views and so on. Academic datasets (including the proposed dataset) do not have large enough samples and are 50/50 balanced. Because of this, it is difficult to understand whether the proposed approach will work correctly with minimal human involvement.

Overall, the article cannot be accepted in its current form and needs improvement. From the scientific side, it does not bring any significant results. On the engineering side, there are doubts about the correct operation of the system in real conditions - additional experiments are needed to determine the reliability of the system.

Comments on the Quality of English Language

In the paper “Efficient Human Violence Recognition for Surveillance in Real-Time” the authors propose a new real-time violence detection model which is designed for real-life scenarios. To evaluate the approach, 4 different datasets were used. The structure is good, the key ideas of the proposed method are clear. The authors also proposed the new dataset VioPeru, which I downloaded and checked for similarities between the train and validation clips, and everything looks fine. Therefore, the proposed dataset is a good contribution to the automatic video surveillance area.

The main doubt about the proposed system is its resistance to false positives. Violence is an extremely rare occurrence, and for real systems it is also important to have a minimum number of false positives.  The authors claimed that they designed it for real-life scenarios, so it needed to be evaluated on several long additional videos without violence. This could be some YouTube videos with street views and so on. Academic datasets (including the proposed dataset) do not have large enough samples and are 50/50 balanced. Because of this, it is difficult to understand whether the proposed approach will work correctly with minimal human involvement.

Overall, the article cannot be accepted in its current form and needs improvement. From the scientific side, it does not bring any significant results. On the engineering side, there are doubts about the correct operation of the system in real conditions - additional experiments are needed to determine the reliability of the system.

Author Response

Thank you very much for taking the time to review this manuscript. Please find the detailed responses and the corresponding corrections in track changes in the re-submitted files.
